# A Review of the Multipronged Attack of Herpes Simplex Virus 1 on the Host Transcriptional Machinery

**DOI:** 10.3390/v13091836

**Published:** 2021-09-14

**Authors:** Thomas Hennig, Lara Djakovic, Lars Dölken, Adam W. Whisnant

**Affiliations:** 1Institute for Virology and Immunobiology, Julius-Maximilians-University Würzburg, 97078 Würzburg, Germany; thomas.hennig@uni-wuerzburg.de (T.H.); lara.djakovic@uni-wuerzburg.de (L.D.); 2Helmholtz Center for Infection Research (HZI), Helmholtz Institute for RNA-Based Infection Research (HIRI), 97080 Würzburg, Germany

**Keywords:** herpes simplex virus, RNA polymerase II, transcription, host shutoff, promoter-proximal pausing, C-terminal domain, polyadenylation, splicing

## Abstract

During lytic infection, herpes simplex virus (HSV) 1 induces a rapid shutoff of host RNA synthesis while redirecting transcriptional machinery to viral genes. In addition to being a major human pathogen, there is burgeoning clinical interest in HSV as a vector in gene delivery and oncolytic therapies, necessitating research into transcriptional control. This review summarizes the array of impacts that HSV has on RNA Polymerase (Pol) II, which transcribes all mRNA in infected cells. We discuss alterations in Pol II holoenzymes, post-translational modifications, and how viral proteins regulate specific activities such as promoter-proximal pausing, splicing, histone repositioning, and termination with respect to host genes. Recent technological innovations that have reshaped our understanding of previous observations are summarized in detail, along with specific research directions and technical considerations for future studies.

## 1. Introduction

Herpes simplex virus type 1 (HSV-1) is the cause of the common cold sore as well as a leading agent in infectious blindness and is capable of establishing severe skin lesions in addition to life-threatening encephalitis. A hallmark of herpes viral infections is a cytopathic lytic phase of viral replication coupled with lifelong, latent infections that periodically reactivate to produce new viral progeny. The lytic phase of HSV infection has long served as a paradigm for how viruses shut down the expression of host genes in infected cells. While host shutoff broadly serves to reroute macromolecular synthesis towards viral replication, antagonizing the array of cellular immune responses is essential for viral spread in vivo. RNA viruses such as influenza and members of the *Alphaviridae,* which encode their own polymerases, have evolved to globally inhibit specific events in host transcription [1], or degrade cellular RNA polymerases directly [2]. HSV, like all other herpes viruses, requires the machinery of the host to express viral RNA. What is unique to HSV, however, is the multitude of cellular transcriptional events the virus antagonizes on host chromatin while simultaneously preserving these activities on viral genes.

Though RNA metabolism in HSV-infected cells has been investigated since the 1950′s [3,4], technological advances in RNA sequencing have revolutionized describing new phenomena in co-transcriptional RNA processing, identifying novel regulatory networks, and elaborating the fates of individual gene products. Other papers have discussed the elements of viral DNA that recruit Pol II and regulate the temporal cascade of viral gene expression [5,6]. Here, we focus on summarizing our current understanding of how HSV disrupts the transcription of host genes in favor of its own gene expression by dissecting major events during RNA biogenesis. These topics include promoter recruitment, promoter-proximal pausing, pre-mRNA splicing, as well as polyadenylation and alterations of the Pol II holoenzyme (Figure 1). Most of the research discussed here has utilized HSV-1 due to increased global prevalence and reduced virulence for staff safety. Still, the high conservation between relevant proteins makes it likely that the phenotypes also occur with HSV type 2, more commonly referred to as genital herpes, though naturally, this is worth experimental confirmation.

## 2. Alterations in RNA Polymerase II Holoenzyme and Activity

Early studies of transcription in HSV-infected cells measuring radioactive nucleotide incorporation observed a ~50% reduction in total RNA synthesis by 4 hours post-infection (hpi) [7,8,9,10,11,12,13,14,15,16,17,18,19], concomitant with a shift of remaining transcriptional activity towards viral DNA. Although this loss of activity eventually occurs with all three mammalian DNA-dependent RNA polymerases, a drastic reduction in transcription of Pol II genes was visible by 3 hpi in nuclear run-on assays [20]. Over the years, a significant body of work demonstrated that regulation of RNA Polymerase (Pol) II and its associated cofactors are essential in expressing viral genes [21,22] and are the primary targets for viral proteins involved in shutting off host transcriptional responses. Evidence that Pol II has different global activity on cellular and viral DNA was observed with robust expression of a β-globin gene with its native promoter inserted into the viral genome, while transcription of the endogenous cellular β-globin was ablated by 5 hpi [23]. HSV facilitates this apparent disparity by sequestering its genome into a selectively accessible viral replication compartment which allows and simultaneously prevents access of certain cellular factors to the viral DNA [24,25].

RNA Pol II is responsible for transcribing all protein-coding and several non-coding RNAs, including lincRNA, miRNA, snoRNA, and multiple snRNAs [26]. In mammals, the Pol II complex is composed of 12 individual protein subunits, while activity is regulated by dozens of additional factors in global and gene-/response-specific fashions. In addition to clarifying details about the general reduction in host gene transcription, high-throughput sequencing techniques have identified that nearly every major step in Pol II transcription of host genes is impacted during lytic HSV infection, as discussed below.

Pol II holoenzymes shift from a ~2MDa fraction by size exclusion chromatography to 670KDa fractions and below in HSV-infected cells, including a loss of TFIIE involved in DNA unwinding at the promoter, in the remaining high molecular weight pool [27]. This study also identified three viral immediate-early proteins—ICP0, ICP4 and ICP27—that co-eluted with Pol II, predominately in the lower molecular weight fractions. Another group confirmed Pol II coprecipitates with ICP27, as was observed for the early gene ICP8 [28]. While these studies demonstrate extensive changes in overall holoenzyme composition, several individual molecular interactions and disruptions on host genes have been described.

The potential to undergo thousands-fold amplification and predominantly nucleosome-free nature of viral DNA during lytic infection may certainly contribute to the sequestration of transcription complexes from host chromatin, but studies performed at time points before DNA replication or with viral mutants have shown that individual viral proteins directly contribute to the loss of Pol II on host chromatin, and in particular gene promoters. While known as both an activator and repressor of viral promoters, immediate-early protein ICP4 was found to deplete Pol II from cellular promoters by as early as 2 hpi using chromatin immunoprecipitation sequencing (ChIP-Seq) [29] and that ICP4 could even be initially recruited to cellular genes. Another ChIP-Seq study identified that ICP4 was responsible for significantly reducing Pol II levels across cellular gene bodies by 4 hpi [30]. Though Pol II has been shown by single-molecule imaging to randomly explore viral DNA in replication compartments (RCs) [24], work with temperature-sensitive mutants suggests that ICP4 and not DNA copy number is what sequesters Pol II and global transcription factors to RCs through ICP4′s interactions with Mediator [29]. Additionally, antisense transcription from host promoters or from within gene bodies gives rise to at least 1000 novel virus-induced cellular transcripts, a subset being activated by transient ICP4 expression [31]. These antisense transcripts, in turn, can regulate the expression of the sense transcript, which may further complicate promoter analysis.

Though it is clear that ICP4 reduces Pol II occupancy on host genes, it remains to be determined at what point transcription factors become unavailable to host genes from the nucleoplasm or recycling from more proximal chromatin. This would clarify activities of viral proteins beyond the competition that inhibit promoter recruitment during the earliest stages of lytic infection or latent reactivation. Another area of interest is exploring how Pol II condensates around sites of active cellular transcription are affected by HSV, either by the action of viral proteins or by physical rearrangement of nuclear structures and host chromatin, as these sites can regulate gene expression in ways not readily apparent by sequencing-based approaches.

## 3. Promoter Clearance and Promoter-Proximal Pausing

Once Pol II has begun transcribing the first few RNA nucleotides (nt), it encounters another regulatory step, promoter-proximal pausing. Pol II and other double-stranded multi-subunit RNA polymerases inherently pause at specific DNA sequences due to physical structures formed by the nascent RNA or DNA-RNA hybrids, termed “intrinsic” pausing (reviewed in [32,33]). Structural studies have demonstrated that a tilted DNA-RNA hybrid exists in paused RPB1, likely formed after the elongated DNA-RNA hybrid translocates in the active site of RPB1 and the DNA template then backtracks without the RNA. This leaves the DNA base within the position to accept incoming NTPs still base paired with the post-translocated RNA in a tilted conformation, unable to accept new NTPs and halting extension of nascent RNA. This conformation can be relieved by structural rearrangements and cleavage of the terminally bonded RNA nucleotide to proceed into elongation. While Pol I and III have their own domains to support these activities, this role for Pol II is filled by RNA-cleavage stimulatory factor TFIIS. Pol II pausing is specifically stabilized within the first 80-nt downstream of the transcription start site of most genes by additional extrinsic factors, referred to here as promoter-proximal pausing (reviewed in [26]).

Promoter-proximal pausing is mainly stabilized by the DRB (5,6-dichloro-1-β-D-ribofuranosylbenzimidazole)-sensitivity-inducing factor (DSIF) and negative elongation factor (NELF) complexes [34]. DSIF, consisting of proteins Spt4 and Spt5, binds the polymerase and clamps upstream DNA and exiting RNA to support proper positioning and retention. NELF, composed of four subunits NELF-A/B/C/E, sits on the edge of the funnel leading to the Pol II active site and requires DSIF to promote pausing by limiting the relative mobility of Pol II modules and physically occluding the binding site of TFIIS. Pausing generally allows for proper recruitment of factors acting later in transcription. In contrast, the release of paused polymerases into productive elongation or premature termination is a major regulatory nexus for viruses like HIV-1 and specific biological processes during development and stress responses [35,36]. The switch from paused to elongating polymerases is typically mediated by positive transcription elongation factor b (P-TEFb), which phosphorylates Pol II, DSIF, and NELF. This leads to the dissociation of NELF and a switch of DSIF from a negative to a positive elongation factor that remains associated with the polymerase and recruits additional downstream factors.

The first study to perform Pol II ChIP-Seq in HSV infection identified a clear loss of promoter-proximal pausing for a subset of 61 cellular genes whose overall occupancy was unchanged by 4 hpi in murine cells [37]. The reduction in pausing was observable for hundreds of additional genes by precision nuclear run-on analysis (PRO-Seq) as early as 3 hpi in human cells [38], as this technique typically has higher signal-to-noise ratios and allows precise mapping of 3′-ends of nascent RNA [39]. Notably, pausing peaks were observed on viral genes using both techniques [37,40], indicating that the factors disrupting pausing—or facilitating its rescue—are not equally active on viral and cellular genes.

The ability of ICP4 to both promote and inhibit the expression of cellular genes has been explored in a recent study centered around pausing [30] where several ICP4-upregulated host genes, which exhibited a reduction in the relative amount of NELF-A and in wild-type HSV infection compared to mock or a ∆ICP4 virus. While this data indicates that ICP4 influences the activity of pausing regulators, other studies suggest viral genes have different requirements for NELF and DSIF. Knockdown of Spt5 severely lowered expression of the viral late gC RNA in one study, while moderate effects were observed for the early ICP8 RNA for both Spt5 and NELF-E knockdown [41]. This study also demonstrated that the viral protein ICP27 coprecipitated with Spt5 in a DRB-responsive manner, while another group further identified ICP22 as a major determinant for Spt5 localization to viral DNA [42]. Spt5 is also copurified with ICP22 in HeLa nuclear extracts [43]. Affected by at least three viral proteins and remaining associated with transcribing polymerases after the pause release, Spt5 would thus be an interesting focus of future work. Determining the associations between Pol II and TFIIS, which relieves the tilted conformation of the paused RNA-DNA hybrid, could also provide mechanistic insights into pausing regulation on the host and viral genes.

In general, these studies highlight the complicated networks that viral and cellular proteins form. Each immediate-early protein mentioned can affect additional pathways that in turn globally influence transcription factors already regulated by other means. One apparent conflict regarding the loss of promoter-proximal pausing is the fact that HSV inhibits P-TEFb kinase activity, a key facilitator of pause release and whose inhibition globally results in increased polymerase pausing [35,36]. More on this is discussed in the next section. It would thus be of benefit to gain detailed structural analyses of individual polymerase complexes or observe individual activities reconstituted in vitro to separate initial causes from downstream effects.

## 4. CTD Phosphorylation

The post-translational modification of the largest Pol II subunit, RPB1, is perhaps one of the most dynamic regulatory events in gene expression and central to numerous transcriptional responses. The most well-studied changes are on the C-terminal domain (CTD), consisting of 52 repeats of the consensus amino acid motif Tyr1-Ser2-Pro3-Thr4-Ser5-Pro6-Ser7 (Y1-S2-P3-T4-S5-P6-S7). Each non-proline residue serves as a site of phosphorylation, which recruits other complexes necessary for proper transcription. Numerous other modifications exist, including site-specific methylation, proline isomerization, glycosylation, and ubiquitination (reviewed in [44]). Many non-consensus heptapeptide repeats, particularly variants in the seventh amino acid position, are enriched in the more distal of the mammalian repeats while other important post-translational modifications occur in regions of RPB1 outside of the CTD [26].

The migration pattern of RPB1 during SDS-PAGE allowed for the identification of a complete loss of the hyperphosphorylated IIo band in HSV infection and replacement with an intermediately phosphorylated band, dubbed IIi, by 5 hpi [45]. This required the de novo expression of viral proteins, particularly ICP22 [46]. Since then, many groups have identified CTD regulation as a major consequence of both host and viral gene expression.

As mentioned in the previous section, polymerases are productively released from promoter-proximal pausing due to the kinase subunit of P-TEFb, cyclin-dependent kinase 9 (CDK9), which phosphorylates DSIF, NELF, and Ser2 of the CTD. It is important to note that other Ser2 kinases exist, and blocking CDK9 can also affect their downstream recruitment [36]. Still, activities of these enzymes have not yet been directly explored in HSV infection. A specific loss of Ser2 phosphorylation (pS2) in infected Vero cells was observed by Western blot, and this required ICP22 [47,48]. Interestingly, work from another group indicated that pS2 is downregulated by ICP27 in HeLa cells [49]. Though these studies varied in cellular contexts and methods such as soluble protein vs. total cell lysates, these findings can be reconciled by the model that ICP22 inhibits CDK9′s ability to phosphorylate Ser2. At the same time, ICP27 promotes the degradation of hyperphosphorylated RPB1. Recently, it was found that phospho-Ser7 (pS7) is down-regulated in addition to pS2 while the other major CTD modifications are preserved. Both ICP22 and ICP27 seemed to contribute to this loss in fibroblasts [50]. This may occur by the same means as the reduction of pS2 in infection, though pS7 loss was not observed with transient expression of ICP22 in HeLa cells [51]. Downregulation of pS7 may result from CDK9 inhibition evolved around pS2. However, there may be other impacts on CTD regulation caused by pS7 loss, as this mark stimulates CDK9 activity on other CTD residues in vitro [52]. Both pS2 and pS7 recruit the Integrator complex to terminate non-polyadenylated transcripts such as Pol II-derived snRNA and replication-dependent histone mRNA, but Integrator has also been shown to be a global regulator of promoter-proximal pausing [53] and is regulated during stress conditions to induce termination defects of mRNA [54]. Thus, clarifying the significance of pS7 loss and Integrator function during infection may illuminate novel regulation.

CDK9 coprecipitates with ICP22 [55] and a short, transiently expressed sequence of ICP22 amino acids 193-256 is enough to inhibit kinase activity [51], while pS2 is retained in viral infection with ICP22 mutants lacking amino acids 240–340, but not 213–240 [56], indicating that this entire region is not necessary for binding CDK9. Interestingly, transiently expressed ICP22 was found on a cellular gene by ChIP, suggesting that the loss of CTD phosphorylation is not a result of failure to recruit CDK9 to sites of transcription [51]. Instead, ICP22 and CDK9 are recruited to sites of transcription, and at least for cellular genes, this leads to a local reduction in pS2 hyperphosphorylation and transcriptional elongation as measured by ChIP qPCR [43,51].

It is important to note that ICP22 has recently been demonstrated to enhance transcriptional elongation on viral genes by PRO-Seq [57], indicating that polymerases can exhibit different activities on cellular and viral chromatin, which may be facilitated through the recruitment of transcriptional Pol II co-factors by ICP22. Furthermore, ICP22 coprecipitates with another Ser2 kinase, CDK12, and numerous other transcription elongation factors, while the functional consequences during infection remain unclear [43]. An additional consideration is that VP16 can coprecipitate with CDK9, which might relieve the inhibitory activity of ICP22 [43,58]. ICP22 has been found to also bind the murine CD80 promoter by ChIP and inhibit its transcriptional activity in vitro and in vivo [59]. CD80 is expressed on the surface of several antigen-presenting cell types and regulates adaptive immune responses in both positive and negative manners through interactions with proteins such as CD28, CTLA-4, and PD-L1. Downregulation of CD80 protein levels is observed specifically for dendritic cells in an ICP22-dependent manner [59].

Mutant viruses lacking ICP22 replicate poorly in murine ocular models and are sensitive to interferon [60]. At the same time, recombinant expression of CD80 or deletion of the aforementioned ligands can directly influence the degree of reactivation from latency and corneal pathology [59,61,62,63,64]. Interestingly, the region of ICP22 responsible for reduced CD80 promoter activity is within amino acids 305-345 [65], well outside of the region binding CDK9, indicating multiple mechanisms by which ICP22 inhibits the transcription of cellular genes.

HSV encodes two viral kinases, US3 and UL13, and both have been implicated in Pol II regulation. Still, it is difficult to distinguish whether they directly target host factors or indirectly regulate them by modifying viral proteins, particularly ICP22. Full induction of the intermediately phosphorylated IIi required both ICP22 and UL13 [66], though both genes supported microscopic colocalization of CDK9 and RPB1 in foci presumed to be replication compartments [55]. UL13 was also found to be required for the localization of ICP22 to RCs [67]. Another study found that coprecipitation of CDK9 with ICP22 required US3, and that US3 actually supported CDK9 phosphorylation of the CTD in vitro [68]. This study also confirmed a role for UL13 in the accumulation of IIi, while observing different requirements for US3 in different cell types. As both kinases directly phosphorylate ICP22 in addition to numerous other proteins, there could be multiple, temporally regulated layers of interactions that facilitate phosphorylation to the IIi form while limiting the accumulation to hyperphosphorylated IIo. Both kinases are packaged into the virions [69], though it is unclear if the incoming amounts are sufficient to influence transcriptional remodeling. Additionally, the formation of both total and pS2 RPB1 foci is affected by the mutation of three phosphorylation sites in ICP27 [70], indicating roles for kinases in the global remodeling of transcription environments.

The maintenance of other CTD marks during infection is likely a consequence of requiring them for transcription of viral genes. Another possibility is the action of viral proteins supersedes their normal roles in transcription and that direct regulation of the CTD mark was not advantageous during evolution. Lacking a clearly defined role, but whose mutation results in a range of defects [71], there is currently little to specifically suggest pY1 is dysregulated in infection or what unique phenotypes would be discernable among the numerous other transcriptional defects. In addition to pS2, cellular transcription termination sites are enriched for phospho-Thr4 (pT4), and it is intriguing to think that HSV could additionally remove pT4 as a means of shutting down host 3′-end formation. However, as discussed below, failure to terminate host mRNAs during infection predominately operates by protein interactions with polyadenylation factors, making it unclear if pT4 is needed for viral genes or if redundant phospho-CTD strategies were simply not adapted to inhibit host mRNA polyadenylation. At the beginning of mRNA transcription, CDK7 phosphorylates CTD Ser5, and this helps recruit capping machinery to nascent RNA. Studies did not observe a decrease in levels of phospho-Ser5 (pS5) by Western blotting [50,72], and ChIP studies using Pol II antibodies that were specific for pS5 or the N-terminus of RPB1 observed similar trends on both the host and viral genome [29,30,42,73]. CDK7 also copurifies with Pol II isolated from infected cells [27], and genes with reduced pS2 in ICP22-expressing cells maintained an equal fraction of pS5 [51]. These data indicate S5 phosphorylation is one of the few processes that does not appear to be disrupted explicitly by HSV, likely being critical in the transcription of viral mRNA.

Additional regulation of factors such as proline isomerization or glycosylation can only be studied indirectly or with specific mass spectrometry-based approaches, and none have yet been attempted in HSV infection [44]. It has been observed that ICP27 facilitates RPB1 ubiquitination during infection, signaling for degradation as a possible means of clearing hyperphosphorylated Pol II [49,74]. Interestingly, these studies found prevention of pS2 loss with proteasome inhibitors. While the majority of pS2/7 could be residing on a small proportion of hyperphosphorylated IIo in the cell, phospho-CTD reductions occur before a comparable drop in total RPB1 protein levels in multiple cell types indicating that remodeling is regulated beyond bulk turnover [47,50]. In contrast to HeLa cells [72], total RPB1 protein levels were stable through the peak of viral transcription in primary fibroblasts, while an almost complete loss was observed by 24 hpi. This loss was partially prevented by mutation of the Lysine 1268 polyubiquitination site, which mediates the proteasomal degradation of RPB1 during transcription-coupled DNA repair. In contrast, mutation of this site had no impact on phospho-serine loss. Reconciling CTD remodeling with RPB1 degradation will provide insight into the temporal regulation of polymerases during infection, as this data suggests that multiple pathways may be involved.

One avenue of interest in this particular regard is Pol II trafficking into virus-induced chaperone (VICE) domains. These domains were named for their localization of multiple cellular protein chaperones, proteins associated with heat shock, ubiquitination, and proteasomal degradation. They are proposed to serve various roles in protein quality control and early replication compartment formation [74,75,76,77,78,79]. Relevant to this discussion is that Pol II can be trafficked to VICE domains where it has been proposed to undergo ubiquitin-mediated degradation [74]. Proteasome inhibition, which prevents pS2/7 loss [49,50], also prevented the formation of these domains [74]. It remains to be determined whether trafficking to VICE domains facilitates CTD remodeling by localizing RPB1 to viral proteins or other cellular factors or if RPB1 is fated only for degradation. VICE domain formation can require viral proteins ICP0, ICP22, or ICP27 in different cell types [60,70,74,75,80], complicating direct correlations to the loss of VICE domains with the different phenotypes of CTD phosphorylation associated with these proteins. Live-cell microscopy studies to measure the rates of RPB1 trafficking between the nucleoplasm, VICE domains, viral replication compartments, and host chromatin could provide valuable insights into mechanisms of global Pol II remodeling and degradation.

## 5. RNA Processing in Splicing and Termination

Sequencing newly synthesized RNA isolated by chemical labeling demonstrated that 3′-end formation and polyadenylation were globally disrupted on most cellular genes [81]. Studies using ChIP-Seq confirmed an increased Pol II occupancy downstream of host genes during HSV infection [37]. Disruption of host Pol II transcription termination (DoTT) turned out to be a major contributing factor to the globally observed shutdown of cellular protein synthesis as these improperly terminated mRNAs are not exported from the nucleus and are thus removed from the translatable mRNA pool. Mammalian cells induce global extensive transcription downstream of genes (DoGs) in response to several abiotic stresses, while a significant overlap exists with the genes exhibiting failure in salt, heat, and HSV infection in fibroblasts; characteristics unique to HSV were identified [82]. The percentage of transcripts on a gene that failed to terminate (upwards of ~70% by 8 hpi vs. ~30% in salt or heat) and the distance they traveled downstream were much greater in HSV infection.

Another study identified that the extended length of polymerases downstream of genes in HSV infection compared to stress was due to the viral protein ICP27, whose transfection was sufficient to inhibit 3′-end formation of many cellular genes [83]. ICP27 was found to directly interact with the Cleavage and Polyadenylation Specificity Factor (CPSF), in a manner that excluded the symplekin protein and prevented 3′-end formation. ICP27 has long been known as a regulator of polyadenylation on viral genes [84], and the presence of an ICP27-binding site proximal to the polyA site was found to rescue 3′-end formation and polyadenylation of viral, and some cellular, mRNA. While the overall tendency for the host is disruption of CPSF function, ICP27-binding results in many alternative polyadenylation sites on cellular genes, typically upstream of usual sequences, and these can indeed be exported and translated [85,86].

The disruption of transcription termination can account for several previously described defects in the processing and expression of host mRNAs during infection, particularly regarding roles assigned to ICP27. The failure of polymerases to properly terminate resulted in elongation into downstream genes, accounting for the apparent induction of hundreds of host mRNAs observed when only considering reads within annotated gene bodies. Interestingly, read-in transcription into downstream genes commonly was accompanied by impaired splicing, indicating that polyA site recognition of the nascent mRNA by cellular factors in the nascent RNA transmits signals to the actively transcribing polymerase that interfere with splicing mechanisms downstream. Many of the defects described in pre-mRNA splicing could thus be only observed in downstream genes read into by termination-incompetent polymerases. In contrast, splicing of the initial upstream gene generally occurred normally. Furthermore, the resulting significant readthrough transcripts are displaced from host chromatin but generally not exported to the cytoplasm. Disruption of transcription termination thereby directly contributes to host shut-off [81,82]. Though technology at the time generally limited the ability to distinguish specific events from broad defects of polymerases failing to terminate, ICP27-mediated regulation of the latter two functions, splicing, and downstream nuclear export have been topics of considerable focus.

Indications that HSV, or ICP27 in particular, can inhibit splicing were observed by host mRNAs migrating at the higher molecular weight on Northern blots [87,88], or studying the expression of spliced reporter plasmids or total RNA levels of individual cellular genes [89,90,91,92,93,94,95,96]. Contemporaneously, studies with the α-globin gene indicated that there may be a separate phenomenon explaining the accumulation of unspliced transcripts from a global inhibition of splicing [97,98]. ICP27 was also observed to coprecipitate with antisera against Sm proteins [99], which bind the 3′-ends of snRNAs and promote spliceosome assembly; with Spliceosome-associated protein 145 (SAP145), which helps tether the U2 snRNP [100], as well as SR protein kinase 1 (SRPK1), which phosphorylates splicing factors [101,102]. The structure of the recently determined ICP27 RGG domain/SRPK1 interaction [103] has revealed that this competitively precludes SRPK1 binding to splicing factor serine/arginine-rich splicing factor 1 (SRSF1). Furthermore, redistribution of splicing complexes in the nucleus to speckles surrounding viral DNA could be observed in infection dependent on ICP27 [92,104,105,106,107,108,109]. These effects, in combination with ICP27′s association with the nuclear pore and export of viral mRNA, previously lead to the conclusion that ICP27 is a major regulator of cellular pre-mRNA splicing and that this inhibition or other mechanisms lead to nuclear accumulation of host transcripts. There is thus a substantial body of evidence that ICP27 can affect splicing factors, even though nascent RNA profiles from different times of infection exclude a global and generalized inhibition of splicing [81].

An important question to resolve is whether intron retention caused by ICP27 stimulates alternative polyA site usage or if ICP27 binding to GC-rich sequences near other polyA sites terminates transcription before proper recruitment of snRNPs and splicing. An additional complication with total RNA analysis arises from the observation that splicing can influence sensitivity to VHS degradation [110]. We emphasize the need for investigations into splicing to use techniques that distinguish RNAs generated at the proper promoter for a gene from those of upstream genes failing to terminate. Such techniques include those using chemical labeling (e.g., 4sU-Seq), long-read sequencing, or minimally normalization of polyadenylated RNA to chromatin-associated rather than total/total nuclear RNA. Chromatin-associated RNA closely matches nascent transcriptomes determined by chemical labeling in HSV infection [111], and is currently the most cost-effective and procedurally simple approach to differentiate co- from post-transcriptional events directly compatible with established HSV infection protocols for total RNA.

## 6. Histone and Chromatin Regulation

Chromatin is a dynamic structure that helps to regulate the accessibility of DNA to transcriptional machinery, thus being closely linked to gene activity [112]. The nucleosome, the basic unit of chromatin, consists of a protein core composed of 147 bp of DNA wrapped around the histone protein octamer (reviewed in [113]). The nucleosome octamer comprises two copies of each of the canonical histones—H3, H4, H2A, H2B—which interact in an ordered manner during the nucleosome assembly. Linker histone H1 plays an essential role in maintaining the higher-order structure of chromatin through locking DNA wrapped around the histone core at the dyad axis. Other reviews discuss the rich topic of chromatin on viral genes during lytic infection and latency [114,115,116]. Here, we focus on the observations for cellular genes.

Host genes exhibited significantly more open chromatin regions (OCRs) downstream of failed polyA sites in HSV infection, but not in salt or heat stress, as measured by assay for transposase-accessible chromatin using sequencing (ATAC-Seq) [82]. This is suggestive of a defect in histone repositioning for elongating polymerases due to the actions of viral proteins. Interestingly, OCRs are exclusively observed downstream of affected polyA sites but not within gene bodies. However, transcription into actively transcribed downstream genes still results in OCR within the respective gene bodies. This indicates that not the nature of the affected chromatin regions but rather signals from the partially recognized polyA site within the nascent mRNA alter the composition of the actively transcribing Pol II and impair histone repositioning in the wake of Pol II.

FACT and SPT6 are among several identified histone chaperones with established roles in nucleosome assembly/disassembly during the Pol II-mediated transcription elongation. FACT is a heterodimeric histone chaperone composed of two subunits, Spt16 (suppressor of Ty 16) and SSRP1 (structure-specific recognition protein 1), which promote transcription elongation through nucleosomes (reviewed in [117]). It was identified that ICP22 interacts with both FACT subunits by Co-IP and mass spectrometry [42,43], while ICP8 is also purified with Spt16 [118]. FACT can act with P-TEFb to alleviate promoter-proximal pausing [119], and promote Pol II elongation through nucleosomes [120]. FACT thus represents another regulatory nexus impacted by HSV worth further investigation.

There is evidence demonstrating histones of host chromatin are broadly affected by HSV infection. All linker histone H1 variants were observed to increase mobilization away from chromatin during infection by fluorescence microscopy in manners independent of ICP0 but enhanced by early viral gene expression [121]. Similar increases in the unbound pool were observed for core histones H2B, H4, H3.1, and variant H3.3 [122,123]. The ChIP-qPCR analysis identified a loss of histone H3 on actively transcribed GAPDH and U3 genes during infection, but not on a non-transcribed pericentric satellite sequence [124]. An increase in the repressive H3K9me3 mark was observed on cellular genes in the presence of immune factor IFI16 [125]. Increased mobility of multiple histones has also been observed with transient expression of ICP4 [126]. Furthermore, dynamic changes in the levels of a wide array of histone post-translational modifications during HSV infection at both total and chromatin-associated protein levels have been identified by mass spectrometry [127]. While the changes in histone locations and modifications can enhance viral gene transcription, additional effects on host gene expression are likely. It would thus be of interest to clarify whether the observed mobilization of linker histones is linked to the observed dOCR formation. Furthermore, it needs to be identified in future ChIP-Seq studies which histones show alterations and to include regions downstream of failed polyA sites rather than limiting the analysis to the areas typically transcribed in uninfected cells.

## 7. Broader Networks and Future Considerations

HSV infection induces a transcriptional program in cells that gears them towards the production of progeny virus particles. While microarray studies indicated the seeming induction of several host genes [128,129,130], particularly antiviral responses, viral and cellular profiles induced are highly variable between individual cells and depend on various factors that include virus dose, virus stock quality, viral genetic variability, the status of the cell and cell type tested. Notably, RNA-Seq data has to be carefully analyzed to avoid misinterpretations as many transcripts are affected by termination failure and, in turn, are atypically spliced, might extend into downstream genes, and, more importantly, fail to get exported to the cytoplasm [81,82,111]. The post-transcriptional stability and, thereby, viral and cellular RNA levels are regulated by the nuclease activity of the aptly named virion host shutoff (VHS) protein, further complicating conclusions of transcriptional regulation from the level of total, cytoplasmic, or polyadenylated RNA. While recent studies have identified many interesting findings from total RNA analysis [131,132,133,134,135], general conclusions at the level of transcription are complicated by the varying technical methods and biological contexts utilized. Interestingly, nuclease activity of VHS globally reduced Pol II transcription of host genes, as was observed before with the SOX nuclease from murine gamma-herpesvirus 68 (MHV68) [111,136]. A proper discussion of host mRNA accumulation, its translation, and resulting outcomes on infection, particularly regarding innate immunity, is outside the scope of this review.

Despite the array of transcriptional and post-transcriptional barriers blocking cellular responses to HSV infection, cellular transcription pathways can be activated. Expression of the embryonic transcription factor double homeobox 4 (DUX4) is induced following HSV-1 infection, and this leads to the accumulation of numerous downstream genes, including antiviral proteins such as TRIM43 [137,138]. Genes repressed by DUX4 induction are significantly enriched in the set of genes transcriptionally downregulated during HSV infection, further identifying DUX4 and possibly other embryonic transcription factors as master regulators during infection [111]. Nascent RNA-Seq analysis identified that only a slight fraction of genes not expressed in uninfected fibroblasts are transcriptionally upregulated in infection, outside of DUX4 genes and those upregulated by type I and II interferons [111]. This study also investigated the role of VHS on transcriptional activity and observed that VHS, through its nuclease activity, caused the downregulation of a set of genes that are associated with the fibroblast lineage (adhesome, ECM organization, metalloproteinases, etc.) and might thus be an important factor in driving the de-differentiation program by destabilizing the mRNAs of certain transcription factors.

Most research on transcription during HSV infection has focused on Pol II, but there is nothing to suggest that this is due to a lack of Pol I or III regulation during infection. As mentioned above, transcriptional activity of all three polymerases decreases within the first few hours after viral entry. Shortly after the development of an antibody against it, the Pol III-associated La protein was observed to relocalize to the cytoplasm as well as the cell surface in HSV infection [106,139,140]. 5S rRNA pseudogene transcripts can regulate immune responses to HSV by relocalizing to the cytoplasm and binding to RIG-I [141]. Pol III has been proposed to facilitate innate immune recognition by transcribing cytosolic DNA [142], and Pol III inhibitors reduced interferon responses to infection with HSV and several other DNA viruses [143]. At the very least, as rRNA levels have served as normalization controls in multiple RNA-Seq and qPCR studies, it would be of benefit to rule out specific downregulation of these genes during lytic infection to better clarify how global RNA synthesis is impacted by metabolic states of infected cells when quantifying disruptions of Pol II-related activities. In addition, HSV-1 induces transcription of telomeric repeat-containing RNA (TERRA) in an ICP0-dependent manner, though the implications of this remain unclear [144]. Last but not least, HSV also impacts mitochondrial gene transcription. The viral UL12.5 nuclease localizes to mitochondria and mediates mitochondrial DNA depletion to interfere with intrinsic defense mechanisms [145,146], and mitochondrial RNAs can regulate immune response to HSV infection [147].

There are many directions to explore in understanding how HSV regulates host transcription. A comprehensive analysis of truly HSV-1-upregulated programs should include the proper omics approaches and selection of genes not affected by read-in transcription from upstream polyA site failure. Isolation of chromatin-associated RNA is a relatively cost-effective method to study transcriptional responses that more closely match the truly nascent profile than total RNA. Established techniques such as ChIP- and mNET-Seq can quantify viral proteins’ impacts on histone repositioning and Pol II CTD regulation. Non-sequencing-based techniques such as high-resolution microscopy can determine the exchange of histones or transcription factors at sites of cellular transcription and even distinguish differences between individual cells. Additional structural information and reconstituted in vitro measurements to study personal activities will help clarify the roles of different viral proteins that converge on individual cellular factors. It is important to emphasize again that the majority of studies discussed in this review utilized HSV-1. While HSV-1 and -2 share considerable conservation at the genomic level, the latter exhibits greater clinical virulence. Unlike HSV-1, which co-speciated with humans, HSV-2 evolved in non-human ancestral hosts. It is thus of interest to determine if variations in homologous viral gene products, particularly in the nuclear immediate-early proteins, affect interactions with cellular transcription factors. These, in turn, can impact the cascade of viral gene expression, host immune responses, and resulting pathology.

Accurately summarizing the array of attacks HSV performs on host transcription, Randall Jay Cohrs (Randy; 1952–2021) at the 2021 Colorado Alphaherpesvirus Latency Society Symposium said, “HSV is like a railroad spike, it’s hard to study an individual process because it breaks everything while VZV [varicella zoster virus, another alphaherpesvirus] is like a nail.” We would like to thank Randy for his friendly personal discussions and innumerable scientific contributions and believe that ongoing research has the potential to shape that railroad spike into a tool that can be manipulated in clinical and oncolytic settings.

## Figures and Tables

**Figure 1 viruses-13-01836-f001:**
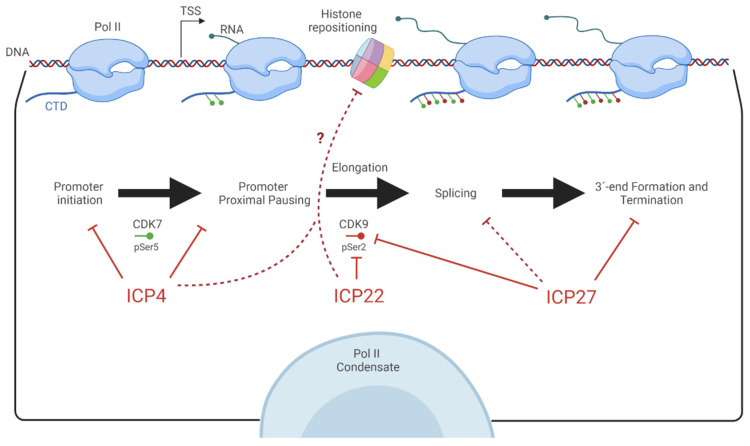
Graphical representation of the described transcriptional events herpes simplex virus antagonizes on host genes. Viral proteins are indicated in red with their reported inhibitory activities discussed in this review marked. The dashed arrows indicate possible links to observed defects in histone repositioning, or reconsiderations of splicing defects based upon new findings.

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
