# Peer review of "A Review of the Multipronged Attack of Herpes Simplex Virus 1 on the Host Transcriptional Machinery"

_viruses, 2021, doi:10.3390/v13091836_

Round 1

Reviewer 1 Report

The authors in this review article extensively describes the role and function of RNA Polymerase (Pol) II in infected cells with HSV. Review focuses on changes in Pol II holoenzymes and post-translational modifications in relation to host gene response. The review is well written, fully comprehensive and a detailed description of the current topic. To study more about the role of ICP22 in relation to its binding with CD80 promotor or the effect of ICP22 absence on the overall impact of host genes after HSV-1 infection is also an important measure to include in this review. Therefore, please cite PMID 34281386, 34287036, 24089574 and 31387116 in relation to this work.

Author Response

We are grateful for the positive comments and for listing these articles which were unfortunately not included in our first submission.  The topic of CD80 promoter regulation by ICP22 is certainly relevant to the theme of this review and we have included discussion of this topic in lines 319-330 of our new submission.

Reviewer 2 Report

In this manuscript, Hennig et al provided an up-to-date review of the various mechanisms of HSV-1 controls the transcriptional apparatus in infected host cells. The review is succinct and well written. Of the two human alphaherpesviruses, HSV-1 has been the most investigated, and though similar in genome composition and organization to HSV-2, it will be interesting to know whether HSV-2 orchestrates other significantly different transcriptional control mechanisms that perhaps explain in part, its greater role in the pathogenesis of genital herpes.

There is just a couple of minor edits for the authors’ consideration.

Line 121: delete “and”

Line 517: change “Randall Jay Cohrs (1952-2021)” to Randall Jay Cohrs (Randy; 1952-2021) or Randall “Randy” Jay Cohrs (1952-2021).

Line 520: “We would like thank Randy” to “We would like to thank Randy

Author Response

We thank the reviewer for their positive feedback.  The idea that variations in how HSV-1 and -2 regulate transcription accounting for differences in virulence is an interesting one, and it is important for the review to re-emphasize that much of this work has been conducted with HSV-1.  As such, we have expanded the discussion in lines 532-539 of our new submission.  We have made the minor edits as suggested.